# Characterization of prevalent genetic variants in the Estonian Biobank body-mass index GWAS

Erik Abner [1,9] ✉, Kanwal Batool [1,9] ✉, Nele Taba [1], Tiit Nikopensius[1], Kristi Läll[1], Anastasiia Alekseienko[1], Anders Eriksson [2,3], Joel Rämö [4], Hele Haapaniemi [4], Hanna Maria Kariis [1], Liis Haljasmägi[5], Urmo Võsa [1], Taavi Tillmann[1,6], Uku Vainik[1,7,8], Kelli Lehto[1], Hanna M. Ollila [4], Kai Kisand [5], Estonian Biobank Research Team* & Tõnu Esko[1]

Population-specific genome-wide association studies can reveal high-impact genomic variants that influence traits like body-mass index (BMI). Using the Estonian Biobank BMI dataset (n = 204,747 participants) we identified 214 genome-wide significant loci. Among those hits, we identified a common non-coding variant within the newly associated *ADGRL3* gene (−0.18 kg/m²; $P = 3.21 \times 10^{-9}$). Moreover, the missense rare variant *PTPRT*:p.Arg1384His associated with lower BMI (−0.44 kg/m²; $P = 2.51 \times 10^{-10}$), while the protein-truncating variant *POMC*:p.Glu206* was associated with considerably higher BMI (+ 0.81 kg/m²; $P = 1.48 \times 10^{-12}$), both likely affecting the functioning of the leptin-melanocortin pathway. *POMC*:p.Glu206* was observed in different North-European populations, suggesting a broader, yet elusive, distribution of this damaging variant. These observations indicate the previously unrecognized roles of the *ADGRL3* and *PTPRT* genes in body weight regulation and suggest an increased prevalence of the *POMC*:p.Glu206* variant in European populations, offering avenues for developing interventions in obesity management.

Obesity is a highly heritable trait[1], resulting from the combined effects of genetic factors and environmental influences. Variants in genes associated with appetite and metabolism can cause obesity[2], and changes in leptin signaling and adipocyte genes are primary contributors to monogenic obesity[3]. Monogenic and polygenic forms of obesity share common biological pathways[4], particularly involving the leptin-melanocortin pathway[5]. Despite the high heritability of obesity, much of its genetic influence remains unexplained, with recent estimates suggesting datasets of up to 50 million participants are needed to fully understand this trait's genetic complexity[6].

Although various methods exist to describe obesity, body mass index (BMI) remains the most straightforward tool to identify overweight and individuals with obesity in a population[7]. BMI-associated susceptibility loci exhibit variable phenotypic effects across different populations due to underlying pathway disparities[8]. Cross-trait meta-analyses reveal that disease patterns can be population-specific, with

[1]Estonian Genome Center, Institute of Genomics, University of Tartu, Tartu, Estonia. [2]Centre for Genomics, Evolution and Medicine, Institute of Genomics, University of Tartu, Tartu, Estonia. [3]Health Data Unit, Applied Research Center, Metrosert, Tallinn, Estonia. [4]Institute for Molecular Medicine Finland (FIMM), Helsinki Institute of Life Science (HiLIFE), University of Helsinki, Helsinki, Finland. [5]Institute of Biomedicine and Translational Medicine, University of Tartu, Tartu, Estonia. [6]Institute of Family Medicine and Public Health, University of Tartu, Tartu, Estonia. [7]Institute of Psychology, University of Tartu, Tartu, Estonia. [8]Montreal Neurological Institute, McGill University, Montreal, QC, Canada. [9]These authors contributed equally: Erik Abner, Kanwal Batool. *A list of authors and their affiliations appears at the end of the paper. ✉e-mail: erik.abner@ut.ee; kanwal.batool@ut.ee

distinct associations between BMI and diseases in different populations[9,10].

Genome-wide association studies (GWASs) have been instrumental in identifying genetic contributors linked to obesity[11]. Since the first reported association between *FTO* and BMI[12,13], additional key susceptibility genes such as *BDNF*, *LEPR*, *PCSK1*, *POMC*, and *MC4R* have been confirmed by GWAS, displaying phenotypic heterogeneity influenced by ethnicity and sex[14,15]. Such variants in or near obesity-related genes can exert subtle modulatory effects on BMI[2], providing insights into the underlying mechanisms of complex traits and guiding the development of targeted therapies[16].

The current GWA study for BMI used the population-specific imputation reference panel data derived from the Estonian Biobank (EstBB) participants[17], with the aim of identifying previously unrecognized loci and protein-structure altering variants associated with BMI. Moreover, we aimed to identify variants that affect the younger population, as EstBB participant age at BMI measurement is on average 13–20 years younger than participants from other large-scale studies focusing on BMI[18–20]. In addition to validating numerous previously known BMI associated loci, we identified coding variants in the leptin-melanocortin pathway, and characterized new moderate-impact variants in the *POMC* and *PTPRT* genes. The protein-truncating *POMC* variant is associated with higher BMI, and the *PTPRT* missense variant is associated with lower BMI, suggesting that *PTPRT* is a possible drug target against weight gain. Overall, our results emphasize the utility of population-specific biobanks in discovering new associations for common complex traits.

## Results

### Population-specific analysis supports prior genetic evidence

We acquired adult BMI values for 204,747 EstBB participants from electronic health records (EHRs) and self-reported questionnaires[21–23]. As EstBB contains multiple BMI values per participant, we used the earliest possible measurement (Supplementary Fig. 1), since genetic effects on body weight are more likely to manifest at an early adulthood age[24,25].

Using genotype data imputed with high-coverage Estonian population-specific reference panel[26], we performed a GWAS on 14,203,082 imputed and genotyped variants. Association analysis was carried out using the REGENIE tool[27], yielding in 214 genome-wide significant loci ($P \leq 5 \times 10^{-8}$). The lead variants replicated numerous previously known BMI-associated loci, with strongest associations being near *FTO*, *MC4R* and *TMEM18* genes (Supplementary Fig. 2). Despite accounting for relatedness, the genomic inflation factor remained high ($\lambda s = 1.639$), which is common in highly powered

quantitative trait GWASs involving related participants[8,28]. However, using linkage disequilibrium score regression (LDSC), we determined that the inflation was primarily driven by polygenicity, leading us to conclude that additional correction was unnecessary (Supplementary Fig. 3).

The BMI data from the EstBB has been utilized in previous GWA studies[14,18]. However, the EstBB has recently undergone data updates quadrupling its participant numbers since 2019. Given that the BMI values were collected from various sources, we employed LDSC on our results to validate our results via genetic correlation with previously published GWAS summary statistics. Among the 1418 summary statistics available from different phenotypes in CTG-VL application[29], the top 33 significant results displayed a substantial genetic correlation ($R_g > 0.6$) with our GWAS results (Bonferroni level of significance at $P < 3.5 \times 10^{-5}$), and were all related to BMI, body weight and adiposity (Supplementary Data 1). The lead result was 'Body mass index (BMI)' from UKBB ($R_g = 0.881$; $P < 1 \times 10^{-300}$), highlighting the validity of the EstBB BMI GWAS results. The heritability estimate for BMI was $h^2 = 0.226$ (SE = 0.012), which is in line with previously published estimates[6,18,28].

### Identification and validation of previously unreported loci

Of the 214 independent loci identified (Supplementary Data 2), nine loci have not been previously associated with BMI (Table 1). All the lead variants from these loci showed at least nominally significant association in the latest available data freeze from the publicly available FinnGen dataset (freeze version 10) for the 'Body-mass index, inverse-rank normalized' trait, suggesting that these single nucleotide variants (SNVs) represent valid hits (Supplementary Fig. 4).

The locus within the Adhesion G protein-coupled receptor L3 (*ADGRL3*) gene (−0.178 kg/m²; $P = 3.21 \times 10^{-9}$) marks for the first time this gene being associated with BMI, body weight, or adiposity. Genome-wide significant SNVs within this locus are located within *ADGRL3* introns and do not show expression quantitative trait locus (eQTL) signals in the eQTLGen and GTEx databases[30,31]. However, as there are no additional genes within ±2 Mb from the lead variant, *ADGRL3* is potentially the causal gene in this locus.

### Identification of protein-structure altering variants

The interpretation of low frequency variants can be challenging due to the lack of validation data from other biobanks. Therefore, we concentrated only on variants resulting in protein structure alterations, which can be interpreted from a biological perspective. The Estonian population-specific imputation panel contains 12,362,035 low frequency variants with minor allele frequency (MAF) > 0.0001 and <0.05

## Table 1 | The list of previously unreported fine-mapped loci

| rsID | Variant (hg38) | AF | INFO | beta (kg/m²) | SE | P value | PIP | Nearest gene |
|---|---|---|---|---|---|---|---|---|
| rs543956628 | 5_157325310_A_G | 0.013 | 0.959 | 0.395 | 0.067 | $4.66 \times 10^{-9}$ | 0.577 | *FNDC9* |
| rs200876443 | 19_49860328_CAG_C | 0.064 | 0.906 | 0.252 | 0.033 | $9.85 \times 10^{-15}$ | 1 | *PTOV1* |
| rs440401 | 16_283017_G_T | 0.503 | 0.901 | 0.104 | 0.016 | $1.31 \times 10^{-10}$ | 1 | *PDIA2* |
| rs61882742 | 11_18264855_A_C | 0.252 | 0.974 | 0.099 | 0.018 | $2.38 \times 10^{-8}$ | 0.727 | *SAA1* |
| rs643066 | 6_165539356_T_C | 0.704 | 0.976 | −0.103 | 0.017 | $8.61 \times 10^{-10}$ | 0.872 | *C6orf118* |
| rs61937587 | 12_39004369_G_A | 0.196 | 0.987 | −0.112 | 0.019 | $4.57 \times 10^{-9}$ | 0.697 | *CPNE8* |
| rs76652221 | 7_24179438_A_G | 0.099 | 0.988 | −0.155 | 0.025 | $1.03 \times 10^{-9}$ | 0.987 | *NPY* |
| rs13124636 | 4_61662453_A_G | 0.069 | 0.989 | −0.178 | 0.03 | $3.21 \times 10^{-9}$ | 0.654 | *ADGRL3* |
| rs142259845 | 17_37854882_G_T | 0.044 | 0.866 | −0.254 | 0.04 | $1.99 \times 10^{-10}$ | 0.92 | *TBC1D3K* |

Of the 214 genome-wide significant loci identified in the Estonian Biobank (EstBB) BMI GWAS, eight previously unreported loci are listed here, sorted in descending order by effect size (beta). Each SNV represents the variant with the highest posterior inclusion probability (PIP) at its locus, as detailed in Supplementary Data 2. Effect sizes (beta in kg/m²), standard errors (SE), and P values are derived from two-sided linear regression using non-RINT BMI data. INFO scores reflect imputation quality. PIP indicates the probability of the variant being causal. P values are exact and unadjusted. The ADGRL3 locus (bolded) has not been previously associated with BMI, body weight, or adiposity.
*RINT* rank-based inverse normal transformation.

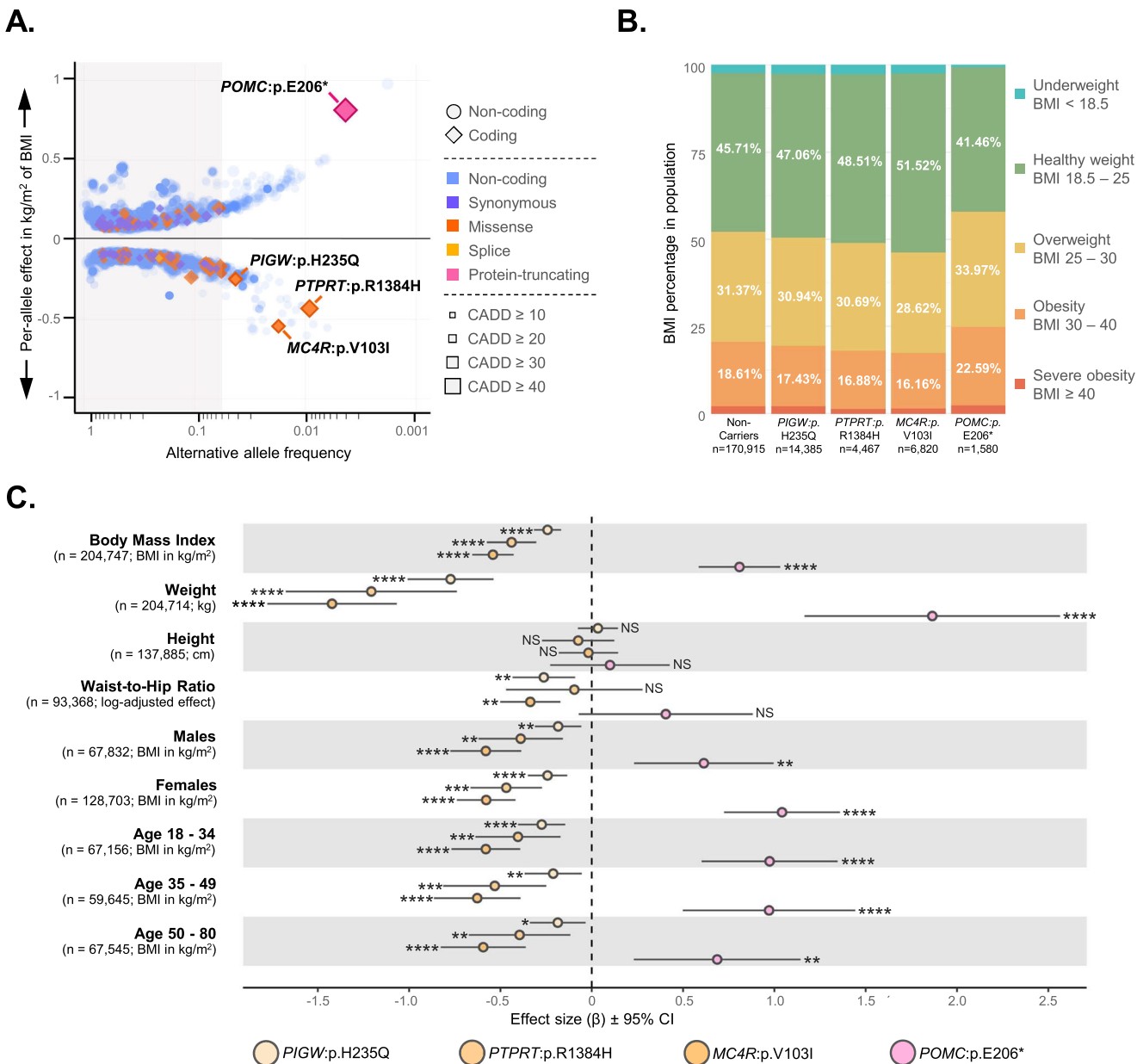

**Fig. 1 | Identification and distribution of BMI-associated rare protein-structure altering variants in EstBB. A** Scatter plot of allele frequencies and BMI GWAS effect sizes for 18,162 significant SNVs ($P < 5 \times 10^{-8}$) from a two-sided linear regression in the Estonian Biobank (EstBB). The x-axis displays allele frequency (log scale) and the y-axis effect size (β in kg/m² from non-RINT GWAS); coding region SNVs are marked as diamonds, non-coding SNVs as circles; point color indicates predicted effect on protein structure; point size represents CADD score-defined deleteriousness. The gray background shading shows AF threshold 0.05 and variants with functional effects on protein structure (AF < 0.05) are annotated. *P* values are exact and unadjusted, and exact values for the highlighted four variants are provided in Supplementary Data 3. **B** Stacked bar chart showing BMI category distribution for carriers of the four protein-structure altering coding variants and non-carriers. BMI categories have been defined according to the World Health Organization guidelines. **C** Forest plot illustrating the non-RINT effect sizes (β) of four coding SNVs on anthropometric traits: body mass index (BMI), weight, height, and waist-to-hip ratio (WHR), including sex- and age-stratified BMI analyses. Points represent the estimated β coefficients from linear regression models comparing heterozygous carriers to non-carriers, adjusted for covariates. Each SNV is color-coded as indicated in the legend. WHR values were log-transformed for visual clarity. Horizontal lines denote 95% confidence intervals. Asterisks indicate statistical significance (*$P < 0.05$; **$P < 0.01$; ***$P < 0.001$; ****$P < 0.0001$; NS = not significant). Exact *P* values and sample sizes per group are provided in Source Data. RINT rank-based inverse normal transformation.

(imputation quality $R^2 \geq 0.8$), which are likely to harbor population-specific and previously unknown variants (Fig. 1A). Filtering for these variants from the GWAS results yielded four implicated variants: *MC4R*:p.Val103Ile ($P = 8.38 \times 10^{-21}$), *POMC*:p.Glu206* ($P = 1.48 \times 10^{-12}$), *PIGW*:p.His235Gln ($P = 3.27 \times 10^{-10}$), and *PTPRT*:p.Arg1384His ($P = 2.51 \times 10^{-10}$) (Supplementary Data 3).

The *PIGW*, *PTPRT* and *MC4R* variants associate with lower BMI and display a higher proportion of individuals in the "Healthy weight" BMI

category and lower proportions in the "Overweight" and "Obese" categories. Conversely, the *POMC* variant associates with increased BMI and the variant shows a lower proportion of individuals in the "Healthy weight" category and considerably higher proportions in the "Overweight" and "Obese" categories, when compared to the non-carrier control population (Fig. 1B).

We investigated the effects of these four SNVs across different age groups, sexes, weight, and height variables (Fig. 1C), using REGENIE to

**A.**

| | | PIGW p.H235Q | PTPRT p.R1384H | MC4R p.V103I | POMC p.E206* |
|---|---|---|---|---|---|
| European | **Overall European** | 0.0104 | 0.0076 | 0.0198 | 0.0007 |
| | Estonian | 0.0435 | 0.0095 | 0.0170 | 0.0035 |
| | Finnish | 0.0261 | 0.0233 | 0.0118 | 0.0016 |
| | Bulgarian | 0.0161 | 0.0030 | 0.0142 | 0.0016 |
| | Other non-Finnish European | 0.0110 | 0.0050 | 0.0190 | 0.0008 |
| | Swedish | 0.0091 | 0.0196 | 0.0260 | 0.0010 |
| | North-western European | 0.0086 | 0.0046 | 0.0198 | 0.0002 |
| | Southern European | 0.0043 | 0.0017 | 0.0102 | 0.0002 |
| | **Admixed American** | 0.0045 | 0.0006 | 0.0033 | 0.0000 |
| | **Ashkenazi Jewish** | 0.0027 | 0.0002 | 0.0039 | 0.0000 |
| | **African/African American** | 0.0014 | 0.0009 | 0.0141 | 0.0000 |
| | **South Asian** | 0.0000 | 0.0004 | 0.0224 | 0.0000 |
| | **East Asian** | 0.0000 | 0.0000 | 0.0218 | 0.0000 |

**B.**

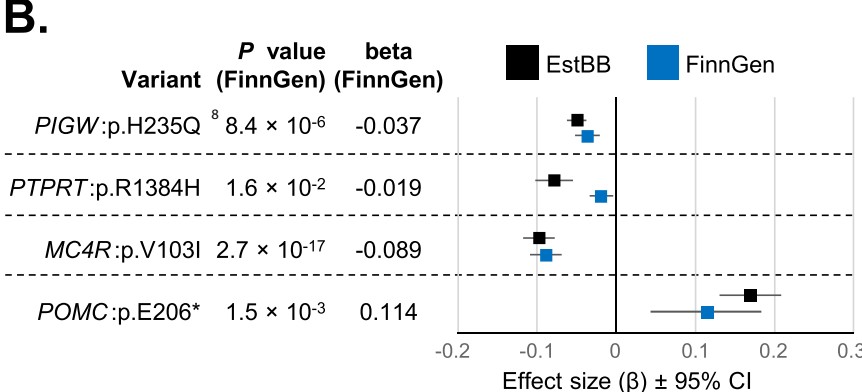

**Fig. 2 | Allele frequencies and validation of coding variants in EstBB and FinnGen populations. A** Minor allele frequencies of the four coding variants across different populations based on the gnomAD database (v2.1.1). Major genetic ancestry groups are highlighted in bold. **B** Validation of the four low-frequency protein-structure altering hits with FinnGen data. Forest plot illustrating the effect sizes (β) of four coding SNVs from FinnGen r10 freeze 'Body-mass index, inverse-rank normalized' from a two-sided linear regression GWAS (blue). The Estonian Biobank (EstBB) BMI GWAS is included as a comparison (black). Betas and *P* values are from rank-based inverse normal transformation (RINT) GWASs; error bars represent ±95% confidence intervals. *P* values are exact and unadjusted. All data represent biological replicates; no technical replicates were used. The EstBB data originate from the same GWAS as presented in Fig. 1. FinnGen GWAS included *n* = 290,820 participants and is publicly available at https://r10.finngen.fi/pheno/BMI_IRN.

conduct linear regression analyses, accounting for relatedness and adjusting for covariates. The impact on BMI was significant across all four SNVs when considering weight, but not height, confirming that the effect on BMI arises from the variation in weight. The effect estimates remained consistent and significant in different age and sex groups, emphasizing the reliability of the individual variant trends. Comparison between *P* values and standard errors (SEs) of older and younger population showed results were reliable for the latter. This implies that genetic effects are more pronounced in younger populations, whereas environmental influences accumulate with age, diminishing the genetic impact and statistical significance.

**Validating the protein-structure altering variants**

We identified notable differences in the frequency of the four protein-coding variants across various populations. According to the gnomAD database, the *MC4R*:p.Val103Ile variant exhibits a MAF in Estonians that aligns with the European average (MAF ~ 0.0170 in Estonians vs. 0.0198 in non-Finnish Europeans). In contrast,

the *PTPRT*:p.Arg1384His variant is more prevalent in Northern Europe, especially among Finns (MAF ~ 0.0233), followed by Swedes (MAF ~ 0.0196) and Estonians (MAF ~ 0.0095). Finally, the *PIGW*:-p.His235Gln and *POMC*:p.Glu206* variants are enriched in Estonians by more than 4-fold, with frequencies of MAF ~ 0.0435 and MAF ~ 0.0035, respectively, when compared to non-Finnish European populations (Fig. 2A). All the four variants are present in the Finnish population, and show nominal association with BMI according to FinnGen r10 freeze (Fig. 2B), supporting their status as valid hits.

Melanocortin 4 receptor (*MC4R*) is a well-established obesity gene, known to be associated with body weight in population-wide studies[32] and with monogenic forms of obesity[33,34]. The *MC4R*:p.Val103Ile variant has previously been shown to be prevalent in Europe[35,36] and has already been extensively studied[37,38]. Our GWAS results demonstrated that *MC4R*:p.Val103Ile decreases BMI by −0.541 kg/m$^2$, which is consistent with previously published studies[28,39]. Therefore, we excluded it from further in-depth analysis.

To our knowledge, the phosphatidylinositol glycan anchor biosynthesis class W (*PIGW*) gene has not been associated with BMI or body weight previously. *PIGW*:p.His235Gln is located in the locus chr17:36482702-36900442 (Supplementary Fig. 5A), which also contains the well-known BMI-associated *GGNBP2* locus[40]. Conditional analysis of the lead hit rs12951387 in this locus decreased the significance of *PIGW*:p.His235Gln below the genome-wide significance threshold (effect size after conditional −0.167 kg/m²; $P = 2.9 \times 10^{-5}$). Although *PIGW*:p.His235Gln retains nominal significance following conditional analysis, it failed to reach Bonferroni correction and was excluded from further in-depth analysis.

The *PTPRT*:p.Arg1384His variant (rs151076965) is the lead hit within its locus, and fine-mapping within the surrounding 1-Mb window identified it as the likely causal variant (PIP = 0.954) among two other significant variants (Supplementary Fig. 5B). The rs151076965 did not colocalize with eQTL signals in the eQTLGen and GTEx databases, suggesting its effect on BMI to result from an alteration in *PTPRT* protein structure or function.

The *POMC*:p.Glu206* variant is within the wider *ADCY3* locus[2,41]. A conditional analysis adjusting for the common lead variant of the locus from the BMI GWAS (rs11676272) showed no significant change in the effect size or *P* value of *POMC*:p.Glu206* (+0.868 kg/m²; $P = 2.86 \times 10^{-14}$), confirming it as an independent hit (Supplementary Fig. 5C). Although *POMC*:p.Glu206* is correlated with the inframe insertion variant rs762710034 (*POMC*:p.Ala201_Gln202insArgAla; LD r² = 1), the insertion occurs in a region of *POMC* that is canonically cleaved and does not encode any functional hormone peptides. This makes the more severe *POMC*:p.Glu206* the more likely causal variant, as it directly impacts the hormone-encoding regions of the gene.

### Characterizing *PTPRT*:p.Arg1384His variant

Protein Tyrosine Phosphatase Receptor Type T (*PTPRT*) had previously not been associated with obesity in population-based genetic studies. The *PTPRT*:p.Arg1384His variant carriers display significantly lower weight among all biobank participants (−1.21 kg; $P = 3.74 \times 10^{-12}$), while self-reported height and waist-to-hip ratios (WHR) did not show any significant association with variant carrier status, suggesting that the effect of *PTPRT*:p.Arg1384His on BMI arose from the weight differences of participants (Fig. 1C, Supplementary Fig. 6A). The independence from WHR changes implies that the variant may influence overall calorie consumption rather than specifically altering body fat distribution.

*PTPRT* is essential for synapse formation, neuronal survival, and synaptic plasticity[42,43]. Phenotypically, *PTPRT* dysfunction has been linked to neurodevelopmental disorders such as autism and intellectual disability[44,45]. Likewise, lower educational attainment is strongly correlated with higher BMI[46,47]. This correlation is also evident in the EstBB (Supplementary Fig. 7), where between the lowest and highest education levels the difference in BMI is at 2.77 kg/m² ($P < 1 \times 10^{-5}$). Although the *PTPRT*:p.Arg1384His variant itself was not correlated with educational status (R² = 0.006; $P = 0.835$), we assessed its effect on BMI after adjusting the GWAS for education and the effect size was −0.391 kg/m² (SE = 0.082), compared to −0.438 kg/m² (SE = 0.069) in the unadjusted analysis. This statistically non-significant change suggests that the modifier effect of *PTPRT*:p.Arg1384His on BMI is largely independent of educational status.

We evaluated the effect of the *PTPRT*:p.Arg1384His variant using clinical data. EHRs are available for each participant since 2004, and can be consolidated into functional biological and clinical PheCode groupings for additional statistical power[48]. To investigate the clinical implications of this variant, we conducted a phenome-wide association study (PheWAS) using logistic regression with REGENIE, accounting for relatedness and adjusting with covariates. The PheCode category 278 "Overweight, obesity, and other hyperalimentation" resulted among the lead hits (OR = 0.890; 95% CI = 0.822-0.963; $P = 3.88 \times 10^{-3}$),

however, none of the PheCode groupings remained significant after Bonferroni correction (Fig. 3A).

The difference from the missense variant might lead to local or global conformational changes in the protein, affecting its overall structure and function. The variant *PTPRT*:p.Arg1384His is considered as possibly pathogenic according to AlphaMissense pathogenicity (0.7077)[49], EVE clinical significance predictor (0.705)[50] and CADD v1.7 deleteriousness scores (34)[51]. The *PTPRT*:p.Arg1384His causes an amino acid substitution in the regulatory phosphatase domain 2 (PTPD2) catalytic site (Fig. 3B)[52,53], which is responsible for interacting with phosphorylated tyrosine (pTyr) residues on other neuronal proteins, such as STAT3[54]. The Dynamut2 protein dynamics model[55] predicts that *PTPRT*:p.Arg1384His mutation results in a stability change of −1.2 kcal/mol, indicating a destabilizing effect on the protein structure (Fig. 3C). The native Arg1384 residue of *PTPRT* forms hydrogen bonds with Leu1275, Pro1341, and Ala1342, contributing to the stability of the PTPD2 catalytic site. In contrast, the Arg1384His variant only forms interactions with Cys1378 (Fig. 3D), a side chain needed for interacting with negatively charged ligands, such as phosphorylated tyrosines[53,56]. Consequently, we infer that the Arg1384His substitution leads to a decreased stability in the catalytic site and might conceivably reduce binding affinity to its ligands.

Moreover, in silico docking hints disrupted binding between pTyr ligand to PTPRT protein (Fig. 3E). The native *PTPRT*:p.Arg1384 forms an ionic bond with the negatively charged $PO_3$ group of the pTyr, while Cys1378 forms hydrogen bonds with the oxygen atoms, thereby stabilizing the ligand. Although docking predictions suggest that the *PTPRT*:p.Arg1384His variant may be capable of similar interactions, the bioinformatic modeling indicates a higher binding free energy (ΔG) between the native PTPRT and pTyr (ΔG: −5.182 kcal/mol vs −4.500 kcal/mol). Therefore, *PTPRT*:p.Arg1384His might destabilize the interactions between PTPRT and its potential binding partners.

### Characterizing *POMC*:p.Glu206* variant

The similarly striking finding from the EstBB BMI GWAS is the variant rs202127120 within the pro-opiomelanocortin (*POMC*) gene, which creates an early protein-truncation codon in the third exon. The rs202127120 has an extreme CADD v1.7 deleteriousness score (37), but is considered by the ClinVar database to be a variant of uncertain significance. The rs202127120 (*POMC*:p.Glu206*) is estimated to increase BMI by +0.809 kg/m² (95% CI 0.585-1.033), has a MAF = 0.0044 among biobank participants, and displays a considerably higher allele frequency in Estonians than in other global populations (Fig. 2A). The variant was originally imputed based on reference panels (INFO = 0.993), and we therefore confirmed the imputation accuracy through Sanger sequencing (Supplementary Fig. 8).

The EstBB recruitment questionnaire asks participants to report their nationality, allowing for the analysis of genetic diversity within the population. Our analysis shows that the *POMC*:p.Glu206* variant has a similar prevalence to Estonians in Baltic populations (Latvians and Lithuanians) and East-Slavic populations, including Ukrainians, Russians, and Belarusians (Fig. 4A). However, according to genetic principal components, Estonians form a separate cluster from these populations, indicating that while the *POMC*:p.Glu206* variant is prevalent in the region, it is distributed across genetically diverse backgrounds (Fig. 4B, Supplementary Fig. 9). Importantly, the effect of *POMC*:p.Glu206* validates nominally according to FinnGen r10 freeze (beta = +0.114; $P = 1.5 \times 10^{-3}$) (Fig. 2B).

The *POMC*:p.Glu206* variant carriers displayed significantly higher weight among all biobank participants (+1.865 kg; $P = 1.72 \times 10^{-7}$) (Fig. 1C, Supplementary Fig. 6A). Moreover, we also witnessed a higher BMI among 18–34 year old participants (+0.974 kg/m²; $P = 2.89 \times 10^{-7}$) (Fig. 1C, Supplementary Fig. 5C) and female participants (+1.042 kg/m²; $P = 1.27 \times 10^{-10}$) (Fig. 1C,

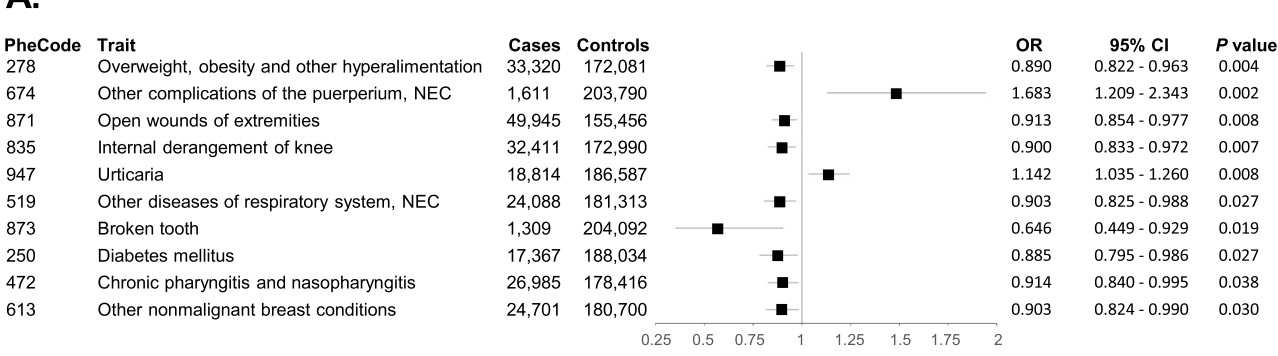

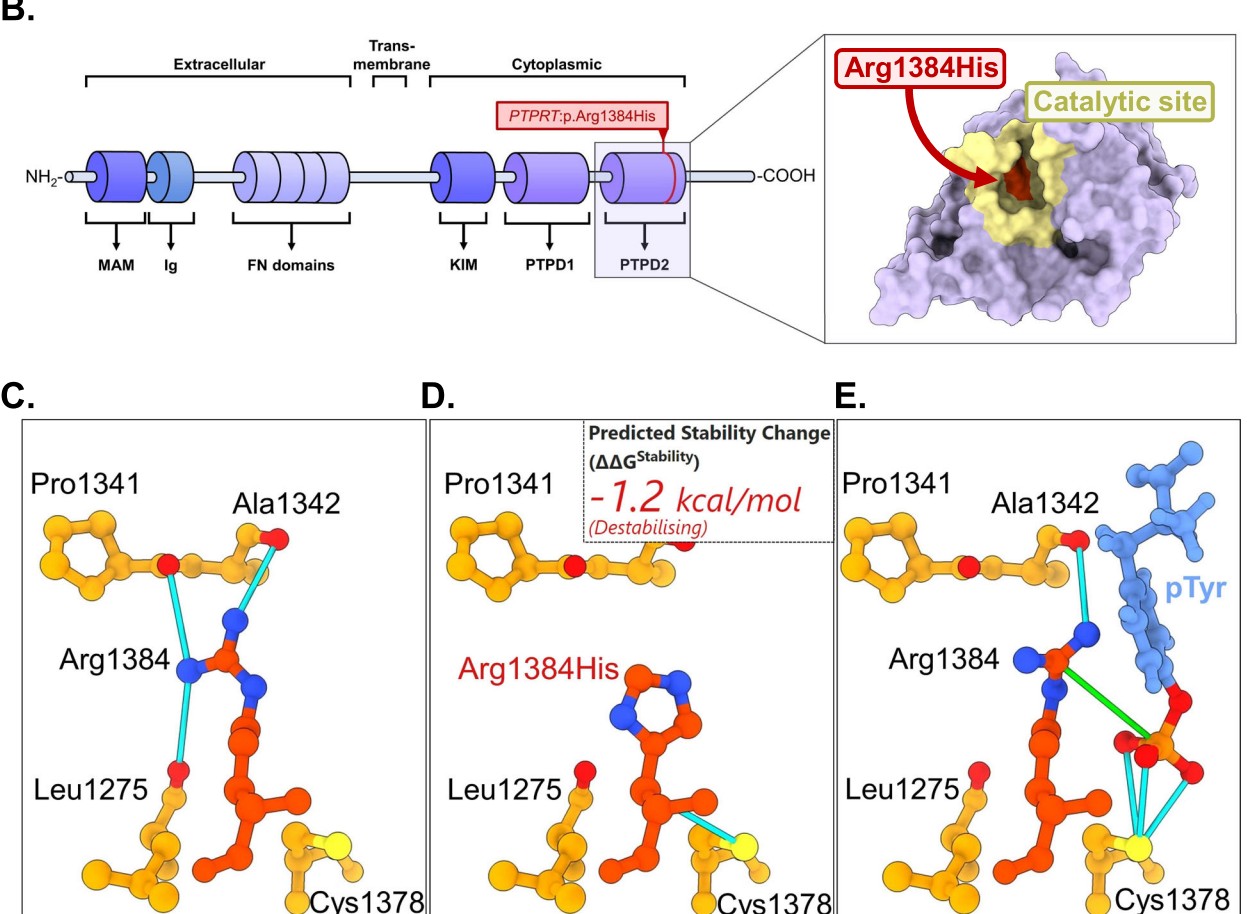

**Fig. 3 | Clinical and structural analysis of *PTPRT:p.Arg1384His* variant. A** EstBB PheCode category-based phenome wide association study (PheWAS) of *PTPRT:p.Arg1384His*, depicted as forest plot. The top 10 associations from an initial exploratory logistic regression-based PheWAS were re-analyzed with REGENIE to account for relatedness and are presented here. Full results from the initial logistic regression analysis are provided in Supplementary Data 4. Points represent odds ratios (OR); error bars indicate 95% confidence intervals (CI). OR odds ratio, NEC not elsewhere classified, CI confidence interval. **B** Minimalistic structure of the PTPRT protein highlighting the location of the missense variant. On the right, the three-dimensional model of domain 2 shows the catalytic site (yellow) and the *PTPRT:p.Arg1384His* variant (red). Protein model based on AlphaFold structure AF-O14522-F1. Visualization created with UCSF ChimeraX[99]. **C** DynaMut2 protein dynamics result showing the native protein structure. Arg1384 forms hydrogen bonds (cyan) with the protein backbone amino acids (light orange). **D** DynaMut2 result with Arg1384His variant. His1384 forms a hydrogen bond (cyan) with Cys1378 instead of the backbone amino acids Leu1275, Pro1341, and Ala1342. **E** In silico ligand docking with native protein. pTyr binds to Cys1378 with hydrogen bonds (cyan) and forms an ionic bond to Arg1384 (green). Docking pose generated using SwissDock[97,98].

Supplementary Fig. 6B). These correlations remained largely unaffected, even after removing participants with diagnostic ICD-10 codes related to severe body morphology changes ($n = 72,756$ participants excluded) from the GWAS analysis (Supplementary Data 5). Analyses using self-reported WHR did not show significant correlation to *POMC:p.Glu206\**, indicating that the variant likely influences calorie intake rather than body fat distribution (Fig. 1C, Supplementary Fig. 6A).

Functionally, the initial POMC peptide prohormone is cleaved into minor polypeptides[2,57], which take part in regulating appetite, pain

## A.

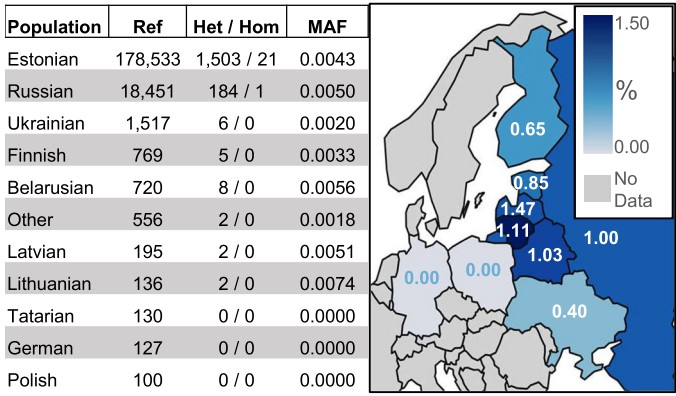

| Population | Ref | Het / Hom | MAF |
|---|---|---|---|
| Estonian | 178,533 | 1,503 / 21 | 0.0043 |
| Russian | 18,451 | 184 / 1 | 0.0050 |
| Ukrainian | 1,517 | 6 / 0 | 0.0020 |
| Finnish | 769 | 5 / 0 | 0.0033 |
| Belarusian | 720 | 8 / 0 | 0.0056 |
| Other | 556 | 2 / 0 | 0.0018 |
| Latvian | 195 | 2 / 0 | 0.0051 |
| Lithuanian | 136 | 2 / 0 | 0.0074 |
| Tatarian | 130 | 0 / 0 | 0.0000 |
| German | 127 | 0 / 0 | 0.0000 |
| Polish | 100 | 0 / 0 | 0.0000 |

## B.

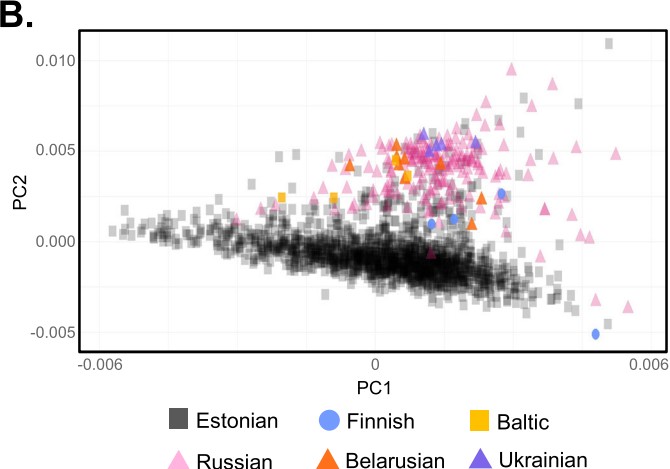

**Fig. 4 | Prevalence of *POMC*:p.Glu206\* variant across nationalities in the Estonian Biobank. A** The carrier numbers and minor allele frequencies of *POMC*:p.-Glu206\* among different nationalities in EstBB are listed in the table. The map illustrates the frequency of *POMC*:p.Glu206\* among individuals based on their self-reported nationality. The percentage of individuals carrying the variant within each nationality group is represented by numbers on the map, with darker shades of blue indicating higher frequencies. Nationalities with no carrier data are marked in gray. Map based on public domain GeoJSON data from the Natural Earth dataset (via github.com/johan/world.geo.json). **B** Principal component (PC) analysis plot of genetic diversity among EstBB participants carrying the *POMC*:p.Glu206\* variant. The Baltic cluster (yellow) contains participants, who reported their nationality to be either Latvian or Lithuanian.

perception, and reward cognition. While regulatory SNVs around *POMC* are also known to correlate to obesity[58], loss-of-function variants are known to cause monogenic obesity[59,60]. The *POMC*:p.Glu206\* stop-gain variant is located in the final exon of the *POMC* gene. Given the Lindeboom-Lehner rules for nonsense-mediated decay (NMD)[61,62], it is likely that the *POMC*:p.Glu206\* variant does not undergo NMD, leading to a truncation that spares the N-terminus but disrupts the regions encoding C-terminus polypeptides (Fig. 5A).

A PheWAS confirmed the enrichment of obesity-related diagnoses among carriers of the *POMC* variant, with the PheCode category 278 "Overweight, obesity, and other hyperalimentation" being the lead hit and surpassing Bonferroni correction (OR = 1.32; 95% CI 1.16–1.50; $P = 2.18 \times 10^{-5}$). It should be noted that the clinical profile of the *POMC*:p.Glu206\* variant carriers do not present other typical *POMC* deficiency symptoms, such as hypopigmentation, hypocortisolism, or hypoglycemia, suggesting that this variant is relatively well-tolerated in the population (Fig. 5B).

## Interaction between polygenic effects and rare variants

Large-scale biobank projects enable detection of additive effects of common and rare variants on various phenotypes. To explore the contributions of low-frequency variants *POMC*:p.Glu206\* and *PTPRT*:p.Arg1384His, we used two previously published polygenic scores (PGSs) from PGS catalog[63] (IDs: PGS002161 and PGS004378) to confirm their effects on BMI (Pearson correlation to EstBB BMI 0.287 vs. 0.224, respectively). We categorized participants from the EstBB into PGS002161 PGS quintiles[64] and assessed the effect sizes of these variants within each quintile. Using linear regression, we evaluated the statistical significance of BMI differences between non-carriers and variant carriers across quintiles. *POMC*:p.Glu206\* showed significant positive associations with BMI, with effect sizes ranging from +0.652 kg/m² (SE = 0.294, $P = 0.0033$) in the lowest quintile to +1.065 kg/m² (SE = 0.317, $P = 7.78 \times 10^{-4}$) in the highest. Conversely, *PTPRT*:p.Arg1384His was significantly negatively associated with BMI from the second to the fifth quintiles, most notably in the fifth quintile (−0.562 kg/m², SE = 0.197, $P = 4.21 \times 10^{-3}$). Based on these results, we can interpret that the effects of the *POMC* and *PTPRT* variants on BMI are independent of the polygenic influences predicted in our BMI analysis (Fig. 6).

## Discussion

Our study using the Estonian Biobank dataset identified nine new common loci significantly associated with BMI, including a protective association within the gene *ADGRL3*. Additionally, we focused on protein-structure altering variants, discovering new moderate-impact variants in *POMC* and *PTPRT* genes with opposing effects on BMI. The *POMC*:p.Glu206\* variant is associated with a considerably higher BMI, while the *PTPRT*:p.Arg1384His variant is linked to lower BMI, suggesting a possible protective role against weight gain.

As the *ADGRL3* locus variant rs13124636 is surrounded by a CTCF site and five distal enhancer-like signatures (±10 kb; Supplementary Fig. 10), the variant could affect proximal gene transcription. Further studies should focus on the function of this intronic site and confirm its effect on *ADGRL3* transcription. The *ADGRL3* gene itself has been previously associated with neurodevelopment, substance abuse, attention deficit hyperactivity disorder and education attainment[65,66]. Intriguingly, ADGRL3 is known to harbor a hormone receptor domain, which has previously been proposed to interact with gastric inhibitory polypeptide (GIP), thereby participating with GIP downstream signaling[67]. *GIP* is a member of the secretin family along with glucagon-like peptide-1 (GLP-1), and is released in response to nutrient ingestion[68]. In addition to increasing insulin secretion and promoting blood glucose regulation, GIP affects appetite regulation, with studies showing GIP receptor knockout mice resisting diet-induced obesity[69,70]. Moreover, the potent anti-obesity drug tirzepatide partially acts as a GIP analog, highlighting the importance of the incretin pathway in weight regulation[71,72]. Therefore, further functional studies focusing on the possible interaction between GIP and ADGRL3 proteins are warranted, with the goal to confirm whether *ADGRL3* plays a role in GIP function and its related metabolic effects.

The *POMC*:p.Glu206\* variant results in an early truncation of the initial protein product and is hypothesized to lower β-MSH and β-endorphin levels in the hypothalamic-pituitary-adrenal axis, leading to an inadequate leptin-melanocortin pathway triggering, and consequently, higher body weight[73]. Our analysis confirms the widespread distribution of *POMC*:p.Glu206\* in the local region, suggesting that Estonians are not the founder population and that undetected carrier populations exist in Northern Europe. To the best of our knowledge, genetic variants within the hormone-encoding regions of *POMC* are rare and typically considered impermissible. *POMC* is considered a gold-standard obesity gene[40], and the loss-of-function *POMC*:p.-Glu206\* variant has previously been described only in single cases with

## A.

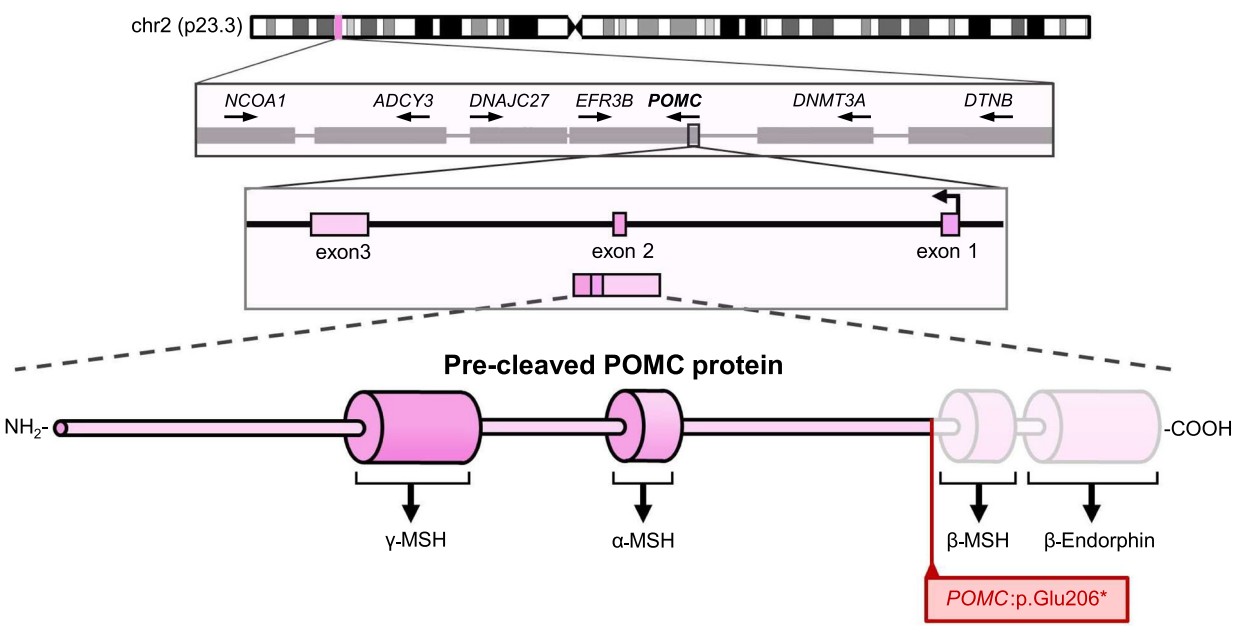

## B.

| PheCode | Trait | Cases | Controls | | OR | 95% CI | P value |
|---------|-------|-------|----------|---|-----|--------|---------|
| 278 | Overweight, obesity and other hyperalimentation | 33,208 | 171,550 | | 1.322 | 1.162 - 1.503 | 2.18×10⁻⁵ |
| 716 | Other arthropathies | 57,325 | 147,433 | | 1.230 | 1.098 - 1.379 | 0.0004 |
| 835 | Internal derangement of knee | 32,320 | 172,438 | | 1.220 | 1.074 - 1.385 | 0.0022 |
| 681 | Superficial cellulitis and abscess | 19,840 | 184,918 | | 1.223 | 1.046 - 1.429 | 0.0115 |
| 524 | Dentofacial anomalies, including malocclusion | 16,344 | 188,414 | | 1.306 | 1.063 - 1.604 | 0.0111 |
| 1000 | Burns | 10,056 | 194,702 | | 1.398 | 1.130 - 1.731 | 0.0021 |
| 916 | Contusion | 81,365 | 123,393 | | 1.135 | 1.031 - 1.249 | 0.0101 |
| 1014 | Effects of heat, cold and air pressure | 4,690 | 200,068 | | 1.693 | 1.241 - 2.311 | 0.0009 |
| 371 | Inflammation of the eye | 29,627 | 175,131 | | 1.166 | 1.021 - 1.331 | 0.0231 |
| 727 | Other disorders of synovium, tendon, and bursa | 33,356 | 171,402 | | 1.204 | 1.061 - 1.365 | 0.0039 |

Odds ratio (95% CI)

**Fig. 5 | Characterization of *POMC*:p.Glu206\*. A** The figure illustrates *POMC* gene's chromosomal location (bolded), exonic structure, pre-protein hormone derivatives, and highlights the impact of the *POMC*:p.Glu206\* variant on POMC pre-protein structure. γ-MSH: Gamma-melanocyte-stimulating hormone; α-MSH: Alpha-melanocyte-stimulating hormone; β-MSH: Beta-melanocyte-stimulating hormone. **B** EstBB PheCode category-based phenome wide association study (PheWAS) of *POMC*:p.Glu206\*, depicted as a forest plot. The top 10 associations from an initial exploratory logistic regression-based PheWAS were re-analyzed with REGENIE to account for relatedness and are presented here. Full results from the initial logistic regression analysis are provided in Supplementary Data 6. Points represent odds ratios (OR); error bars indicate 95% confidence intervals (CI). OR odds ratio, NEC not elsewhere classified, CI confidence interval.

severe early-onset obesity[59,74]. Our results suggest that similar protein-truncating variants could be more common. Analogously, a near-identical and surprisingly common stop-gain mutation exists in the *Pomc* gene of labrador retrievers, where the Asp217\* variant is associated with increased appetite, weight, food motivation and obesity[75,76]. Given the considerable effect of the *POMC*:p.Glu206\* variant, carriers with excessive body weight might benefit from therapeutic approaches targeting food motivation. Long-term clinical trials with GLP-1 receptor agonists have proven sustainable[77,78], conceivably providing variant carriers as a therapeutic option. Drugs like semaglutide and tirzepatide partially reduce body weight through the leptin-melanocortin pathway and could be an effective, widely available option for heterozygous *POMC*:p.Glu206\* carriers. More severe cases and homozygous carriers could be treated with more intrusive therapies, such as the peptide-based MC4R agonist setmelanotide[79] or bariatric surgery. However, further studies will be needed to determine the efficacy of such approaches, as the chronic shortage of β-MSH and β-endorphin due to the protein-truncating variant might hamper the effect of these interventions.

*PTPRT* is predicted to be a loss-of-function intolerant gene (pLi = 1.00), and has not been previously linked to body weight in human studies. However, *Ptprt* knockout mice are resistant to obesity due to decreased *Npy* transcription, resulting in reduced food intake[80]. Canonically, PTPRT protein acts as an upstream negative regulator of the STAT3 transcription factor[54,81], which in turn regulates appetite-controlling genes *NPY* and *POMC*[82–84]. The pseudophosphatase domain of PTPRT sequesters phosphorylated STAT3[52], which could inhibit STAT3 downstream mechanisms from leptin in the context of obesity[85,86]. We hypothesize that the *PTPRT*:p.Arg1384His variant is deficient at sequestering STAT3, thereby potentiating the transcription of its target genes (Fig. 7). If PTPRT participates directly in the leptin-melanocortin pathway, it could be a potential drug target, as protein tyrosine phosphatase family members are considered druggable[87].

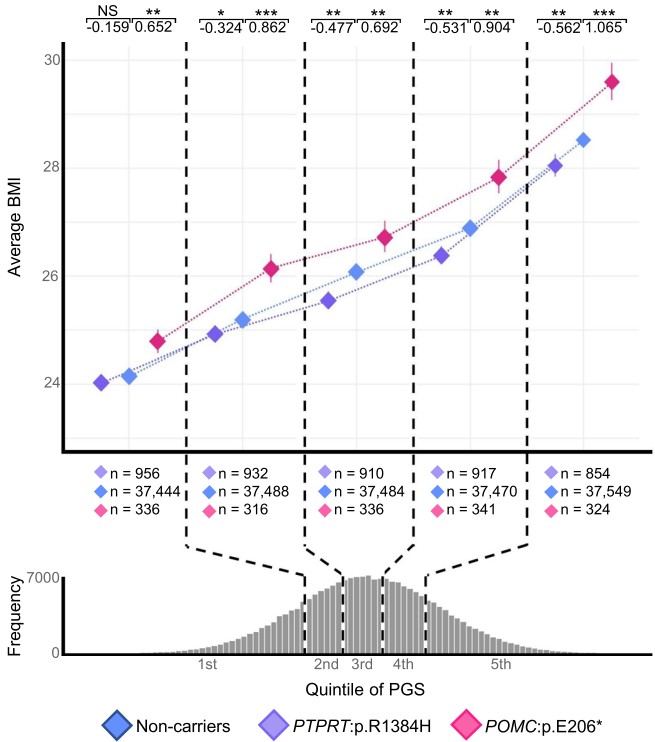

**Fig. 6 | Additive effects of *PTPRT:p.Arg1384His* and *POMC:p.Glu206*\* variants on BMI across PGS quintiles.** In the upper panel, the mean body mass index (BMI) and the standard errors of non-carriers (blue) and carriers of *PTPRT:p.Arg1384His* (purple) and *POMC:p.Glu206*\* (pink) variant carriers within each PGS (polygenic score) quintile group are plotted. The sample sizes for each quintile group, both for non-carriers and variant carriers are indicated below the x-axis. The histogram at the bottom of the figure illustrates the distribution of the PGS for BMI. Within each quintile, we carried out a logistic regression to determine if variant carriers displayed a different BMI from non-carriers from the same quintile as the reference group. The effect sizes and *P* values are displayed above the corresponding quintile group. *P* values are presented as follows: NS (not significant), *\*P* < 0.05, *\*\*P* < 0.01, *\*\*\*P* < 0.001. Exact *P* values are provided in Source Data.

Our study has several limitations. As this is an exploratory study, we did not account for medical conditions during the first stage of analysis, which may introduce bias and affect the subsequent GWAS results. Moreover, while BMI is considered to be a risk factor for numerous diseases, the rarity of the *POMC*:p.Glu206* and *PTPRT*:p.Arg1384His variants creates the issue of statistical power and limits our ability to draw definitive conclusions about the clinical effects of such variants. This could be addressed by a thorough clinical data meta-analysis with other biobanks having carriers of the same variants, such as the FinnGen project[19]. To draw definitive conclusions about these biological effects, it is essential to map other poorly genotyped populations, particularly those in Northern and Eastern Europe, where these variants may be more prevalent. Additionally, while our observational study provides both genetic and bioinformatic evidence, experimental validation is essential to confirm the causal impacts of the *ADGRL3* and *PTPRT*:p.Arg1384His variants. Finally, this work is limited by its consideration of possible confounding environmental influences, such as diet or lifestyle, as they cannot be evenly controlled in a retrospective study.

This study demonstrates the use of a younger biobank cohort to yield robust signals for quantitative traits like BMI, where genetic effects are less confounded by environmental factors or chronic diseases (Fig. 1C). The newly identified non-coding variant in the ADGRL3 locus warrants functional studies to validate the gene's regulatory role in obesity. Moreover, protein-altering variants in the melanocortin-

leptin pathway genes *POMC*, *MC4R*, and potentially *PTPRT*, significantly associated with BMI. Given the pathway's role in appetite regulation[88,89], future research should assess the efficacy of interventions like GLP-1 or GIP receptor agonists in these variant carriers. Such studies could improve our understanding of therapeutic mechanisms of these drugs and advance personalized medicine by identifying effective treatments tailored to different genetic backgrounds.

## Methods

### Cohort description

The Estonian Biobank is a volunteer-based biobank with 212,955 participants in the current data freeze (2024v1). All biobank participants have signed a broad informed consent form and information on ICD-10 codes is obtained via regular linking with the national Health Insurance Fund and other relevant databases, with the majority of the electronic health records having been collected since 2004[21]. Objective measurement values were acquired from electronic health records and self-reported questionnaires. The earliest BMI measurement from adulthood (age ≥18) was used if multiple BMI values were available. Participants with BMI values below 15 kg/m$^2$ and above 70 kg/m$^2$ were excluded from the analyses.

### Genotyping and imputation

All EstBB participants have been genotyped at the Core Genotyping Lab of the Institute of Genomics, University of Tartu, using Illumina Global Screening Array (versions 1, 2 and 3). Samples were genotyped and PLINK format files were created using Illumina GenomeStudio v2.0.4. Individuals were excluded from the analysis if their call-rate was <95%, if they were outliers of the absolute value of heterozygosity (>3 SD from the mean) or if sex defined based on heterozygosity of X chromosome did not match sex in self-reported phenotype data[26]. Before imputation, variants were filtered by call-rate <95%, Hardy-Weinberg equilibrium *P* value < 1 × 10$^{-4}$ (autosomal variants only), and minor allele frequency <1%. Genotyped variant positions were in build 37 and were lifted over to build 38 using Picard. Phasing was performed using the Beagle v5.4 software[90]. Imputation was performed with Beagle v5.4 software (beagle.22Jul22.46e.jar) and default settings. Imputation was done in batches of 5000. A population specific reference panel consisting of 2695 whole genome sequenced samples and standard Beagle hg38 recombination maps were used for imputation. Based on principal component analysis, samples that were not of European ancestry were removed. Duplicate and monozygous twin detection was performed with KING 2.2.7[91], and one sample was removed out of each pair of duplicates.

### Genetic analyses

Genetic association analysis in Estonian Biobank was carried out for all variants with an INFO score >0.4 using the additive or recessive models as implemented in REGENIE v3.2 with standard quantitative trait settings[27]. INFO score filter was increased to ≥0.8 for variants with MAF < 0.01. Two sided linear regression in REGENIE was carried out with adjustment for trait measurement age, age squared, sex, year of birth and 10 principal components as covariates, analyzing only variants with a minimum minor allele count of 10. Two sets of BMI GWAS analyses were conducted. First, the exploratory GWAS utilized rank-based inverse normal transformation (RINT) to identify the most likely true positive hits. Then, for the already identified hits, realistic effect sizes were obtained while rerunning the analysis without using RINT. The Manhattan plot was created using the topr R package[92]. LDSC analyses were carried out with CTG-VL online tool[29]. Variant effects were predicted using the ProtVar[93] and VEP tools[94].

### Other statistical analyses

For results depicted in Fig. 1B, C, Fig. 3A, Fig. 5B, Supplementary Figs. 6A–C and Fig. 6, homozygous variant carriers and multiple

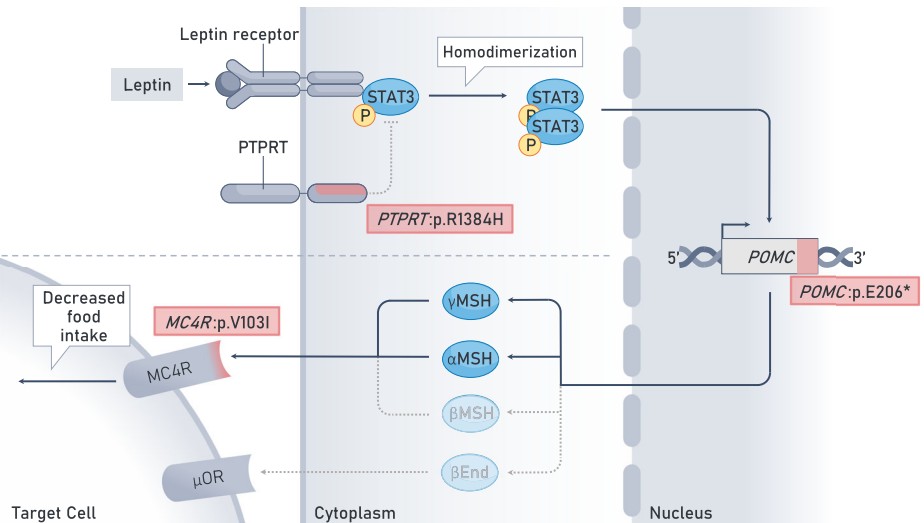

**Fig. 7 | Proposed mechanisms of *POMC:p.Glu206\*, PTPRT:p.Arg1384His* and *MC4R:p.Val103Ile* as part of the melanocortin-leptin pathway.** As part of the leptin-melanocortin pathway, POMC-neurons can be triggered by leptin, a hormone which is produced as a result of food consumption. Upon triggering the leptin receptor, a cascade follows which phosphorylates the cytosolically sequestered STAT3. Canonically, upon phosphorylation, STAT3 homodimerizes, migrates to the cell nucleus and activates its target genes, such as *POMC*. As *POMC:p.Glu206\** results in an early stop codon, and variant carriers will not be able to produce the hormones from the C-terminus, which in turn leads to diminished MC4-receptor and μ-opioid receptor triggering. Finally, *MC4R:p.Val103Ile* has been shown to potentiate the activating signal of paraventricular nucleus neurons. Spotted arrows represent inhibited pathways. PTPRT receptor-type tyrosine-protein phosphatase T, STAT3 signal transducer and activator of transcription 3, POMC pro-opiomelanocortin, MC4R melanocortin 4 receptor, μOR μ-opioid receptor.

variant carriers were excluded. For Fig. 1C and Supplementary Fig. 6A, weight, height, and waist-to-hip ratio were acquired from both self-reported questionnaires of EstBB and medical records. *P* values for these figures were obtained by using two-sided linear regression in REGENIE, accounting for relatedness and adjusting for necessary covariates—age, age squared, sex (except in male and female specific analyses), and year of birth. Other statistical analyses were carried out with R version 4.2.1. PheWAS analyses (Figs. 3A, 5B) were conducted using disease status data derived from electronic health records. Logistic regression was used to estimate associations, adjusting for age, age squared, sex, year of birth, and the first ten genetic principal components. Initial exploratory analyses included related individuals; the top 10 associations were subsequently re-evaluated using REGENIE to account for familial relatedness. FinnGen data for validation analyses was obtained from publicly available summary statistics, data freeze 10.

### Fine-mapping and novel locus identification

For fine-mapping, we employed a custom in-house bioinformatics pipeline, accessible at https://github.com/urmovosa/FinemapAbf/tree/main. This pipeline uses the *finemap.abf* function from the coloc v5.2.3 R package for Approximate Bayes Factor fine-mapping, operating within a 1 Mb genomic window and incorporating a pre-filtering process based on a MAF threshold of 0.01[95]. The identification of novel loci utilized an in-house custom pipeline, which integrates known genetic associations from the GWAS Catalog (v1.0) and the OpenTargets Genetics Portal (v2d version). Employing an Estonian Biobank-specific LD reference panel, we refined SNP selection by excluding highly linked variants within the same 1 Mb window, using an $R^2$ threshold of 0.2 and a window size of 250 kb to ensure the retention of only independent SNVs. The pipeline further identified proxy SNVs for each lead variant based on their physical proximity and LD. For validation study using FinnGen data, we only included novel loci where the lead hit had posterior inclusion probability (PIP) above 0.5.

### Protein structure analysis

Protein structure analysis was conducted using several computational tools. Due to the absence of a previously published crystal structure of PTPRT phosphatase domain 2 (PTPD2), we used the structure published by AlphaFold (ID AF-O14522-F1)[96]. Subsequently, protein dynamics were evaluated using DynaMut2[55] to assess the impact of the Arg1384His mutation on the structural stability of the PTPD2. For docking analysis between the PTPD2 and pTyr, the SwissProt docking function[97,98] was utilized. The analysis focused on identifying the orientation with the lowest docking score where the phosphoryl group of pTyr was correctly positioned within the active site pocket, as previously described[53]. Visualization of the protein structures and docking results was performed using ChimeraX[99].

### Polygenic score analysis

We assessed two PGSs from the PGS Catalog[63] (PGS catalog IDs PGS002161 and PGS004378), ultimately selecting a score developed using the UK Biobank dataset[64]. The PGS002161 score did not include the variants of interest from *POMC* and *PTPRT* genes, nor participants from the EstBB, ensuring independence in our analysis. A Pearson correlation analysis showed a correlation of 0.287 with BMI in our cohort, indicating a considerable association. The chosen PGS was based on a comprehensive set of 990,022 variants, with only two variants not matching those in the EstBB imputation panel. The calculation of the PGS was conducted using the "pgsc-calc" tool developed by the PGS Catalog Team, available at https://github.com/PGScatalog/pgsc_calc.

### Ethics

The activities of the EstBB are regulated by the Human Genes Research Act, which was adopted in 2000 specifically for the operations of EstBB. Individual level data analysis in EstBB was carried out under ethical approval 1.1-12/624 from the Estonian Committee on Bioethics and Human Research (Estonian Ministry of Social Affairs), using data according to release application 3-10/GI/16856 from the Estonian Biobank.

**Reporting summary**

Further information on research design is available in the Nature Portfolio Reporting Summary linked to this article.

## Data availability

GWAS summary statistics are available on GWAS Catalog (https://www.ebi.ac.uk/gwas/home) under accession numbers GCST90624699; GCST90624700; GCST90624701;GCST90624702; GCST90624703; GCST90624704; GCST90624705. The procedure to access EstBB individual-level data has been described at https://genomics.ut.ee/en/content/estonian-biobank#dataaccess, queries can be addressed to Dr Lili Milani (lili.milani@ut.ee). For validation with FinnGen data freeze 10 was used (https://r10.finngen.fi/). Source data are provided with this paper.

## Code availability

All software programs used for the analyses described in this paper are freely available online: REGENIE v3.2 (https://github.com/rgcgithub/regenie); UCSC Genome Browser for SNV analysis (http://genome.ucsc.edu/); Statistical analysis were carried out using R version 4.2.1; FUMA v1.4.0 (https://fuma.ctglab.nl/); LDSC v1.0.1 (https://github.com/bulik/ldsc); PLINK2 (www.cog-genomics.org/plink/2.0/); protein dynamics were evaluated with DynaMut2 (https://biosig.lab.uq.edu.au/dynamut2/); visualization of the protein structures and docking results using ChimeraX v1.8 (https://www.rbvi.ucsf.edu/chimerax/); Fine-mapping (https://github.com/urmovosa/FinemapAbf/tree/main); PGS Catalog Calculator v2.0.0-beta.3 (https://github.com/PGScatalog/pgsc_calc); For FinnGen, code to perform GWAS analyses is available at the FinnGen GitHub (https://github.com/FINNGEN/). Individual plots were created using R v3.6.3, v4.2.2 and v4.3.2, including the R packages ggplot2, RColorBrewer, geojsonio, sf and topr. Map in Fig. 4a was generated using publicly available GeoJSON data from the Natural Earth dataset. The final figures were edited with Microsoft Powerpoint.

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

## Acknowledgements
We want to acknowledge the participants of the Estonian Biobank for their contributions. The Estonian Genome Center analyses were partially carried out in the High Performance Computing Center, University of Tartu. This work was written at writing retreats and writing days organized by the Institute of Genomics, University of Tartu. The Estonian Biobank Sanger sequencing analyses were carried out by the Core Facility of Genomics and the Biobank Lab of the Estonian Biobank. The Estonian Biobank Research Team was responsible for data collection, genotyping, QC, and imputation and consisted of Andres Metspalu (andres.metspalu@ut.ee), Mait Metspalu (mait.metspalu@ut.ee), Lili Milani (lili.milani@ut.ee), Reedik Mägi (reedik.magi@ut.ee), Tõnu Esko (tonu.esko@ut.ee), Mari Nelis (mari.nelis@ut.ee), and Georgi Hudjashov (georgi.hudjashov@ut.ee). The Health Informatics Research Team was responsible for data harmonization, mapping to OMOP CDM, fact extraction of laboratory measurements and consisted of Raivo Kolde (raivo.kolde@ut.ee), Sven Laur (sven.laur@ut.ee), Sulev Reisberg (sulev.reisberg@ut.ee), and Jaak Vilo (jaak.vilo@ut.ee). We want to acknowledge the participants and investigators of the FinnGen study. The FinnGen study is a large-scale genomics initiative that has analyzed over 500,000 Finnish biobank samples and correlated genetic variation with health data to understand disease mechanisms and predispositions. The project is a collaboration between research organizations and biobanks within Finland and international industry partners. The work of the Estonian Genome Center, University of Tartu was funded by the European Union through Horizon research and innovation program under grants no. 810645 (A.E), 894987 (E.A), 101060011 (K.L), 101080117 (A.E), 101117251 (U.Va), 101137201 (E.A.) and 101137154 (E.A.), and through the European Regional Development Fund project MOBEC008 (A.E). Views and opinions expressed are however those of the author(s) only and do not necessarily reflect those of the European Union or European Research Executive Agency. Neither the European Union nor the granting authority can be held responsible for them. Additional funding was obtained from Estonian Research Council Grants PSG809 (T.T), PSG615 (K.Le, H.M.K), PRG1291 (E.A., U.Võ., T.E), PRG1117 (K.K, L.H), PRG1911 (K.L), PRG1414 (N.T), and Center of Excellence of Well-Being Sciences grants no. TK214 (EstBB Research Team) and TK218 (T.T., K.Le, U.Va) by the Estonian Ministry of Education and Research.

## Author contributions
E.A., K.B., T.T., U.Va., and T.E. designed the study. The EstBB Research Team collected and provided the genotype and phenotype data used in the EstBB cohort-level analyses. E.A., K.B., N.T., H.M.K., A.E., J.R., H.H., and L.H. performed the experiments and analyses. T.N. carried out Sanger sequencing data analysis. K.L. calculated the polygenic risk score for EstBB participants. U.Võ. and A.A. developed fine-mapping and variant identification pipelines. K.K., H.O., K.Le., and T.E. supervised the study. E.A. and K.B wrote the first draft of the manuscript. All authors critically reviewed the manuscript.

## Competing interests
The authors declare no competing interests.

## Additional information

## Estonian Biobank Research Team

Andres Metspalu[1], Mait Metspalu[1], Lili Milani[1], Reedik Mägi[1], Tõnu Esko[1], Mari Nelis[1] & Georgi Hudjashov[1]

