## [Transparent Peer Review file · Nature Communications]

Characterization of prevalent genetic variants in Estonian Biobank body-mass index GWAS

Corresponding Author: Dr Erik Abner

Version 0:

Reviewer comments:

Reviewer #1

(Remarks to the Author)

Overall

This is a very interesting and thorough GWAS for BMI using the Estonian Biobank. Several new signals are detected and explored further, with an emphasis on protein-coding variants. I think this is suitable for publication once the minor comments below are addressed.

Specific Comments:

- Title is misleading as it only mentions POMC. This is a GWAS and this should be clear from the title. Equally, it is not clear that the POMC variant is associated with increased BMI.
- It should be noted in line 64 that, apart from MC4R, these genes are not considered to contribute significantly to common obesity, only to rare monogenic forms.
- Protein structure analysis for PTPD2 seems very weak as it is all based on a prediction of the structure by AlphaFold. While this is not unreasonable the authors should be very wary about making specific claims about the effects of residue changes as they have no actual data, only a prediction. I would like to see a more cautious phrasing of any results from this analysis.
- It would be better to see some discussion of how the authors will follow up the ADGRL3 signal, given it is strongly associated with protection but intronic.
- The last two paragraphs contain quite general discussion and should be combined into a smaller final summary paragraph
- For the data availability, will the authors be making the PheWAS data available as well?

Minor comments:

- Publications :
 - o Why isn't Loos and Yeo used as the main review of genetics of obesity early in the introduction? Ref 2 is relatively outdated compared to the recent review and it is already in the reference list so the authors are aware of it.
 - o Heritability and BMI – This is a better reference, though the authors may have missed this as it is very recent. Chodick et al (2024) JAMA Network Open.2024; 7(6):e2419029. doi:10.1001/jamanetworkopen.2024.19029

Reviewer #2

(Remarks to the Author)

The GWAS study of Abner and Batool et al. used data from the Estonian Biobank and identified eight new loci associated with BMI while also validating several previously known BMI-associated loci. Among these new loci, one in the ADGRL3 gene was associated with lower BMI, marking the first time this gene was linked to body weight and adiposity. They further concentrated on the identification of variants in the coding regions potentially altering protein structure and highlighted four variants within the MC4R, POMC, PIGW, and PTPRT genes. They extensively characterized two variants of them in the POMC and PTPRT gene in depth for potential functional relevance using in silico tools. The authors discussed how these coding variants might influence the leptin-melanocortin pathway, a critical regulator of appetite and energy balance. They also proposed potential therapeutic implications, suggesting PTPRT as a possible drug target, and highlighting the potential benefits of GLP-1 receptor agonists for individuals with the POMC variant. The results were validated using external datasets, including the FinnGen biobank. The manuscript is well-written, with comprehensive analyses and conclusive

findings. However, some minor points should be addressed prior publication.

Validity

They demonstrated the robustness of their data by validation with external datasets at multiple steps, and the validation of several known variants further supports the reliability of their analysis procedures. Additionally, they confirmed the POMC:p.Glu206* variant by Sanger sequencing. Therefore, I do not question the validity of their data analysis and the conclusions they draw are conclusive. However, as the authors already mention, results are based only on in silico analyses and need further validation for functional relevance by in vitro and/or in vivo studies. This should be highlighted more in the discussion.

Significance

Although a GWAS with BMI is not novel, they successfully identified previously unknown loci associated with BMI, particularly ADGRL3, which had not been linked to obesity before. These findings are highly significant for the field of obesity research, contributing to the development of new drug targets and paving the way for personalized treatment options.

Data and methodology

The results are reasonable, and the authors presented the results conclusively. Limitations of the study (for example the high genomic inflation factor) are discussed and addressed in their analysis. However, from my point of view it would be reasonable to adjust the level of significance in the GWAS to a more stringent p-value, since the number of tests by imputation is much higher in this study.

Suggested improvements

1. I generally miss the aim of the study. A paragraph about the study rationale would be helpful. I just find a short statement on this in line 69-70.
2. The authors write that BMI (line 204) and WHR (line 257) are largely self-reported. Can the authors exclude that the method of evaluation of BMI or WHR has an impact on the results? Furthermore, was sex also self-reported? This should be addressed in a short limitation paragraph, which is absolutely missing so far.
3. Further, the authors use a lot of abbreviation which are not general terms for non-genetic experts, like HWE (Hardy-Weinberg-Equilibrium).
4. I understood that the authors want to focus on young people to see the genetic effects, but on the other hand they also test effects by socio-economic status – which is to my understanding an environmental condition? This feels a bit controversial at this point. Is this the socio-economic status of the parents or of the included individuals? Why did the authors not consider late onset effects driven by environmental conditions in general, especially if such data is available?
5. Numbering of Figures/Tables: In Line 238 they refer to Table 2, which does not exist. Maybe Figure 2 was meant. Furthermore, Supplementary Figure 5 comes after Supplementary Figure 9. The order should be checked.
6. In line 251-259 the authors write that participants with pregnancy related ICD-10 codes were removed. But it's not clear what the rationale is behind testing/adjusting for this.
7. Although the sex difference in the effect of the POMC protein-truncating variant on BMI is not statistically significant, there is a trend indicating a greater effect in women, as the authors note. Furthermore, despite not being significant, there is a trend suggesting a WHR reducing effect, despite the increase in BMI. It would be interesting to see, if the effect on WHR becomes significant after grouping by sex. Furthermore, the variant was described to be associated with lipedema in two other publications (PMID: 35207755; PMID: 38407896). It's a health condition mainly affecting women, is highly underdiagnosed and frequently misclassified as obesity. The WHR in these patients tends to be low despite having a high BMI. Based on this, I disagree with the conclusion of the authors (line 258/259) and suggest that this variant is affecting body fat distribution. I think it might be relevant to discuss this.

Clarity and context

The text is well written, and the analytical approach is comprehensible described.

References

The authors prove the validity of their results with appropriate literature.

Reviewer #3

(Remarks to the Author)

Version 1:

Reviewer comments:

Reviewer #1

(Remarks to the Author)

I am happy with the author's responses to my comments

Reviewer #2

(Remarks to the Author)

The response to my questions and comments was satisfactory. I have no further comments.

Reviewer #3

(Remarks to the Author)

Response to reviewers

We would like to thank all the reviewers for their kind and helpful review of our manuscript. We have addressed each question raised by the reviewers, modified the manuscript accordingly and find that this has improved the manuscript noticeably.

The reviewers' comments are provided below in black and our response to each point in **blue**. We have made small edits in the manuscript where we found minor errors, or a few important points were missing. Most importantly, we also have made modifications based on the comments of the reviewers. Changes in the manuscript are highlighted in **yellow**.

REVIEWER COMMENTS

Reviewer #1 (Remarks to the Author):

Overall

This is a very interesting and thorough GWAS for BMI using the Estonian Biobank. Several new signals are detected and explored further, with an emphasis on protein-coding variants. I think this is suitable for publication once the minor comments below are addressed.

Specific Comments:

- Title is misleading as it only mentions POMC. This is a GWAS and this should be clear from the title. Equally, it is not clear that the POMC variant is associated with increased BMI.

We thank the reviewer for feedback regarding the title. We understand the importance of ensuring that the title accurately conveys the scope of the study.

To address this, we have revised the title to: "*Characterization of prevalent genetic variants in Estonian Biobank body-mass index GWAS*". This updated title explicitly highlights the genome-wide approach and emphasizes the focus on BMI-associated variants.

- It should be noted in line 64 that, apart from MC4R, these genes are not considered to contribute significantly to common obesity, only to rare monogenic forms.

Since the first GWAS identified the association between *FTO* and BMI, subsequent GWA studies have uncovered or reinforced other key susceptibility genes. We wish to emphasize here the fact that well known susceptibility genes have also been identified by GWAS, thereby demonstrating the usefulness of a GWAS-based approach.

We rephrased the sentence as such: "*Since the first reported association between *FTO* and BMI, additional key susceptibility genes such as *BDNF*, *LEPR*, *PCSK1*, *POMC*, and *MC4R* have been confirmed by GWAS...*" (lines 64-67).

- Protein structure analysis for PTPD2 seems very weak as it is all based on a prediction of the structure by AlphaFold. While this is not unreasonable the authors should be very wary about making specific claims about the effects of residue changes as they have no actual data, only a prediction. I would like to see a more cautious phrasing of any results from this analysis.

We thank the reviewer for the comment and agree completely that bioinformatic predictions have to be approached extremely cautiously. We therefore rephrased that section of the text:

“Consequently, we infer that the Arg1384His substitution leads to a decreased stability in the catalytic site and might conceivably reduce binding affinity to its ligands.” (lines 229-231)

“Although docking predictions suggest that the PTPRT:p.Arg1384His variant may be capable of similar interactions, the bioinformatic modeling indicates a higher binding free energy (ΔG) between the native PTPRT and pTyr (ΔG : -5.182 kcal/mol vs -4.500 kcal/mol). Therefore, PTPRT:p.Arg1384His might destabilize the interactions between PTPRT and its binding partners.” (lines 235-239).

- It would be better to see some discussion of how the authors will follow up the *ADGRL3* signal, given it is strongly associated with protection but intronic.

We appreciate the reviewer’s interest in further discussion regarding the *ADGRL3* signal, particularly its strong protective association. However, as we're witnessing here a novel association, we currently lack substantial BMI-related published data for clear-cut conclusions. *ADGRL3* is strongly expressed in neurons, however it is also present in other tissues, which makes predictions using existing data extremely unspecific (<https://gtexportal.org/home/gene/adgrl3>).

To investigate this signal more comprehensively, separate wet lab experiments are required, which extend beyond the scope of our current study. These studies should focus on evaluating the previously proposed *ADGRL3*-GIP interaction, starting with standard methods, such as co-immunoprecipitation; pull-down assays, co-localization microscopy, etc. If the interaction is confirmed, this would place *ADGRL3* into the framework of GLP-1/GIP pathways.

Otherwise, the *ADGRL3* role in body weight would have to be determined via standard knock-down or knock-out studies.

As for the manuscript, we added a supplementary figure about *ADGRL3* locus and added the following details into discussion:

*“As the *ADGRL3* locus variant rs13124636 is surrounded by a CTCF site and five distal enhancer-like signatures (± 10 kb; Suppl. Figure 10), the variant could affect proximal gene transcription. Further studies should focus on the function of this intronic site and confirm its effect on *ADGRL3* transcription.” (lines 305-308).*

- The last two paragraphs contain quite general discussion and should be combined into a smaller final summary paragraph.

We thank you for the suggestion. We have sharpened the final paragraph as recommended. However, based on feedback from other reviewers, we extended the penultimate paragraph to discuss the limitations of our study. As a result, we have rephrased and refined both paragraphs accordingly:

*“Our study has several limitations. As this is an exploratory study, we did not account for medical conditions during the first stage of analysis, which may introduce bias and affect the subsequent GWAS results. Moreover, while BMI is considered to be a risk factor for numerous diseases, the rarity of the *POMC*:p.Glu206* and *PTPRT*:p.Arg1384His variants creates the issue of statistical power and limits our ability to draw definitive conclusions about the clinical effects of such variants. This could be addressed by a thorough clinical data meta-analysis with other biobanks having carriers of the same variants, such as the FinnGen project. To draw definitive conclusions about these biological effects, it is essential to map other poorly genotyped populations, particularly those in Northern and Eastern Europe, where these variants may be more prevalent. Additionally, while our observational study provides both genetic*

and bioinformatic evidence, experimental validation is essential to confirm the causal impacts of the ADGRL3 and PTPRT:p.Arg1384His variants. Finally, this work is limited by its consideration of possible confounding environmental influences, such as diet or lifestyle, as they cannot be evenly controlled in a retrospective study.

This study demonstrates the use of a younger biobank cohort to yield robust signals for quantitative traits like BMI, where genetic effects are less confounded by environmental factors or chronic diseases. The novel non-coding variant in the ADGRL3 locus warrants functional studies to validate the gene's regulatory role in obesity. Moreover, protein-altering variants in the melanocortin-leptin pathway genes POMC, MC4R, and potentially PTPRT, significantly associated with BMI. Given the pathway's role in appetite regulation, future research should assess the efficacy of interventions like GLP-1 or GIP receptor agonists in these variant carriers. Such studies could improve our understanding of therapeutic mechanisms of these drugs and advance personalized medicine by identifying effective treatments tailored to different genetic backgrounds." (lines 354-376).

- For the data availability, will the authors be making the PheWAS data available as well?

Yes, the PheWAS results can now be found in supplementary tables 4 and 6.

Minor comments:

- Publications :

o Why isn't Loos and Yeo used as the main review of genetics of obesity early in the introduction? Ref 2 is relatively outdated compared to the recent review and it is already in the reference list so the authors are aware of it.

After revisiting the Loos and Yeo review, we realized that while we initially included content from that paper, we opted for a combination of references (ranging from older to more recent) to support the same ideas. However, we have updated the reference to the more recent and authoritative Loos and Yeo review, given its current relevance and coverage of genetic findings in obesity.

o Heritability and BMI – This is a better reference, though the authors may have missed this as it is very recent. Chodick et al (2024) JAMA Network Open.2024; 7(6):e2419029. doi:10.1001/jamanetworkopen.2024.19029

We appreciate the reviewer for bringing this recent reference to our attention. We have now incorporated the Chodick et al. study into the revised manuscript to strengthen our discussion on BMI heritability.

Reviewers #2 & #3 (Remarks to the Author):

The GWAS study of Abner and Batool et al. used data from the Estonian Biobank and identified eight new loci associated with BMI while also validating several previously known BMI-associated loci. Among these new loci, one in the ADGRL3 gene was associated with lower BMI, marking the first time this gene was linked to body weight and adiposity. They further concentrated on the identification of variants in the coding regions potentially altering protein structure and highlighted four variants within the MC4R, POMC, PIGW, and PTPRT genes. They extensively characterized two variants of them in the POMC and PTPRT gene in depth for potential functional relevance using in silico tools. The authors

discussed how these coding variants might influence the leptin-melanocortin pathway, a critical regulator of appetite and energy balance. They also proposed potential therapeutic implications, suggesting PTPRT as a possible drug target, and highlighting the potential benefits of GLP-1 receptor agonists for individuals with the *POMC* variant. The results were validated using external datasets, including the FinnGen biobank. The manuscript is well-written, with comprehensive analyses and conclusive findings. However, some minor points should be addressed prior publication.

Validity

They demonstrated the robustness of their data by validation with external datasets at multiple steps, and the validation of several known variants further supports the reliability of their analysis procedures. Additionally, they confirmed the *POMC*:p.Glu206* variant by Sanger sequencing. Therefore, I do not question the validity of their data analysis and the conclusions they draw are conclusive. However, as the authors already mention, results are based only on in silico analyses and need further validation for functional relevance by in vitro and/or in vivo studies. This should be highlighted more in the discussion.

Significance

Although a GWAS with BMI is not novel, they successfully identified previously unknown loci associated with BMI, particularly *ADGRL3*, which had not been linked to obesity before. These findings are highly significant for the field of obesity research, contributing to the development of new drug targets and paving the way for personalized treatment options.

Data and methodology

The results are reasonable, and the authors presented the results conclusively. Limitations of the study (for example the high genomic inflation factor) are discussed and addressed in their analysis. However, from my point of view it would be reasonable to adjust the level of significance in the GWAS to a more stringent p-value, since the number of tests by imputation is much higher in this study.

Suggested improvements

1. I generally miss the aim of the study. A paragraph about the study rationale would be helpful. I just find a short statement on this in line 69-70.

We thank the reviewer for pointing this out. To address this, we have added a sentence elaborating the study rationale to the introduction section for improved clarity: *"Moreover, we aimed to identify variants that affect the younger population, as EstBB participant age at BMI measurement is on average 13-20 years younger than participants from other large-scale studies focusing on BMI"*. (lines 73-75)

2. The authors write that BMI (line 204) and WHR (line 257) are largely self-reported. Can the authors exclude that the method of evaluation of BMI or WHR has an impact on the results? Furthermore, was sex also self-reported? This should be addressed in a short limitation paragraph, which is absolutely missing so far.

We thank the reviewer for highlighting these points, as self-reported data can oftentimes indeed cause issues [PMID:39695248]. We will address the points raised by the reviewers separately:

- Regarding the effect of self-reported BMI, we conducted additional analyses to address the potential impact of data source differences (self-reported vs. electronic health record-derived). We observed no significant differences in effect sizes between the two sources (please see figure below). While minor differences in P-values were noted, these were expected due to a reduction in case numbers, as BMI

data is not always available from both sources for every individual. This information has been added to Supplementary Table 5, and highlighted in the manuscript (lines 84-85, 383-384).

- For WHR source in EstBB, it is derived exclusively from self-reported data. We have now explicitly clarified this in the manuscript (lines 192 ,266 & 419).

- In response to the question about sex, we clarify that sex information was first obtained from self-reported data and then verified using genotyped data based on X chromosome heterozygosity, as described in the methods section (lines 390-393). Participants with discrepancies between self-reported sex and genetically inferred sex, or those with abnormal chromosomal findings, were excluded from the analysis.

- As per request, the limitations section can now be found in the penultimate paragraph of the discussion:

“Our study has several limitations. As this is an exploratory study, we did not account for medical conditions during the first stage of analysis, which may introduce bias and affect the subsequent GWAS results. Moreover, while BMI is considered to be a risk factor for numerous diseases, the rarity of the POMC:p.Glu206 and PTPRT:p.Arg1384His variants creates the issue of statistical power and limits our ability to draw definitive conclusions about the clinical effects of such variants. This could be addressed by a thorough clinical data meta-analysis with other biobanks having carriers of the same variants, such as the FinnGen project. To draw definitive conclusions about these biological effects, it is essential to map other poorly genotyped populations, particularly those in Northern and Eastern Europe, where these variants may be more prevalent. Additionally, while our observational study provides both genetic and bioinformatic evidence, experimental validation is essential to confirm the causal impacts of the ADGRL3 and PTPRT:p.Arg1384His variants. Finally, this work is limited by its consideration of possible confounding environmental influences, such as diet or lifestyle, as they cannot be evenly controlled in a retrospective study.”*

3. Further, the authors use a lot of abbreviation which are not general terms for non-genetic experts, like HWE (Hardy-Weinberg-Equilibrium).

We appreciate the reviewer’s feedback regarding the use of abbreviations. To enhance clarity for all readers, we have ensured that all abbreviations, including 'Hardy-Weinberg equilibrium' (HWE), are spelled out in full at their first mention. We have carefully reviewed the manuscript to confirm that full forms are provided where necessary.

4. I understood that the authors want to focus on young people to see the genetic effects, but on the other hand they also test effects by socio-economic status – which is to my understanding an environmental condition? This feels a bit controversial at this point. Is this the socio-economic status of the parents or of the included individuals? Why did the authors not consider late onset effects driven

by environmental conditions in general, especially if such data is available?

Addressing the reviewer's question regarding the inclusion of socio-economic status (SES).

- Education has a strong genetic component. The analysis focusing on the lowest and highest education levels was necessary to confirm the existence of this correlation in Estonian Biobank (Supplementary Figure 7), which to our knowledge hasn't been shown with this specific dataset before. *PTPRT* has been associated with neurodevelopment, which could explain its effect on BMI via education. Having established the education-BMI correlation in the dataset, we then wished to observe, whether the *PTPRT* variant could effect education independently. As *PTPRT* variant doesn't correlate with education, we concluded that education was not a confounding factor in the *PTPRT*-BMI correlation.

- The educational data was obtained only from the participants, not from their parents. We acknowledge that we do not have parental education data, which limits our ability to fully disentangle individual and parental SES contributions. Additionally, the environmental data available in the Estonian Biobank is of relatively poor quality, which restricts its use for investigating late-onset environmental effects on BMI.

- Our study remains focused on identifying early genetic factors influencing BMI in Estonian (primarily) working age population.

5. Numbering of Figures/Tables: In Line 238 they refer to Table 2, which does not exist. Maybe Figure 2 was meant. Furthermore, Supplementary Figure 5 comes after Supplementary Figure 9. The order should be checked.

We thank the reviewer for bringing this to our attention. We have corrected the reference in (Line 247) to refer to Figure 2a instead of Table 2. Additionally, we have reviewed and reorganized the numbering of the supplementary figures to ensure they are presented in the correct order.

6. In line 251-259 the authors write that participants with pregnancy related ICD-10 codes were removed. But it's not clear what the rationale is behind testing/adjusting for this.

We thank the reviewer for raising this important point.

Our previous experience using the biobank BMI data has shown us that diagnostic codes related to pregnancy have an effect on objective BMI measurements due to gestational weight gain and obesity during pregnancy [PMID:31987757]. Moreover, these diagnoses are the most common phenotypes that severely affect body morphology in the biobank. As we witnessed a more prominent effect from the *POMC* variant on females in an age-related manner, we decided to use pregnancy as the main argument.

However, we now have addressed this issue by broadening the phenotype exclusions by removing participants with any diagnostic code that could affect body morphology directly (incl. pregnancy, congenital disorders, severe neurological disorders, amputations etc.)

The list of such traits was obtained from a previously published article [PMID: 32442405]. These results have also been added to the manuscript:

"These correlations remained largely unaffected, even after removing participants with diagnostic ICD-10 codes related to severe body morphology changes (n = 72,756 participants excluded) from the GWAS analysis (+0.800 kg/m²; P = 3.22 × 10⁻⁸)" (line 263-265).

7. (a) Although the sex difference in the effect of the *POMC* protein-truncating variant on BMI is not statistically significant, there is a trend indicating a greater effect in women, as the authors note. Furthermore, despite not being significant, there is a trend suggesting a WHR reducing effect, despite the increase in BMI. It would be interesting to see, if the effect on WHR becomes significant after grouping by sex.

We thank the reviewer for this insightful suggestions. We decided to break the question into two segments.

First, we tested the effect of the *POMC* variant on WHR in a sex-stratified manner using linear regression, with age, age-squared, year of birth, and principal components included as covariates. However, we did not observe any significant differences in the effect size between sexes (please see the bar plot below).

(b) Furthermore, the variant was described to be associated with lipedema in two other publications (PMID: 35207755; PMID: 38407896). It's a health condition mainly affecting women, is highly underdiagnosed and frequently misclassified as obesity. The WHR in these patients tends to be low despite having a high BMI. Based on this, I disagree with the conclusion of the authors (line 258/259) and suggest that this variant is affecting body fat distribution. I think it might be relevant to discuss this.

We thank the reviewer for those references, as we were not aware of the possible lipedema connection beforehand. Unfortunately, the Estonian Biobank doesn't have lipedema classified in any specific manner for more detailed analysis. We therefore opted for in-direct analyses, hoping to validate the previously published target gene panel results.

Adjusting WHR with BMI should phenotypically separate fat distribution from general adiposity [PMID:28073971]. We did not witness any significant effect of WHRadjBMI for the *POMC* variant on female participants ($\beta = -0.0004$; SE = 0.0054; $P = 0.882$). However, the variant does increase both hip ($\beta = 1.681$; SE = 0.765; $P = 1.67 \times 10^{-5}$) and waist circumference ($\beta = 1.806$; SE = 0.868; $P = 4.57 \times 10^{-5}$) simultaneously, and at relatively similar rates. If the variant would affect regional fat accumulation we would anticipate a more prominent difference between the two variables, as lipedema causes an increase in hip circumference, but would leave waist circumference relatively unaffected. It should be noted that lipedema self-reported age-of-onset overlaps with our cohort mean age (79.6% of cases report having lipedema before the age of 40) [PMID: 36401222]. As the mean age in our dataset is 43.6

years, it should be representative enough to witness this effect.

Based on these results, it seems challenging to conclude that the *POMC* variant is a risk factor for changes in regional adiposity in the EstBB. Instead, we would like to reiterate our initial interpretation: "... the variant may influence overall calorie consumption rather than specifically altering body fat distribution." (lines 265-267)

Clarity and context

The text is well written, and the analytical approach is comprehensible described.

References

The authors prove the validity of their results with appropriate literature.

Reviewer #3 (Remarks to the Author):

Other minor changes carried out by the authors:

- Minor corrections in figures (axis in supplementary Figure 6; *P* values in Figures 3 and 5; Tabel 1 was re-formatted from image into table format)
- Added Supplementary Figure 10 with *ADGRL3* genomic locus; and Supplementary tables 4-6
- Added additional authors, who helped during the revision process with FinnGen BMI data